# ESD Ideas: Climate tipping is not instantaneous – the duration of an overshoot matters

Paul D. L. Ritchie[1,2], Chris Huntingford[3], and Peter M. Cox[1,2]

[1]Department of Mathematics and Statistics, Faculty of Environment, Science and Economy, University of Exeter, North Park Road, Exeter, EX4 4QE, UK
[2]Global Systems Institute, Faculty of Environment, Science and Economy, University of Exeter, North Park Road, Exeter, EX4 4QE, UK
[3]UK Centre for Ecology and Hydrology, Wallingford, OX10 8BB, UK

**Correspondence:** Paul D. L. Ritchie (Paul.Ritchie@exeter.ac.uk)

**Abstract.** Climate Tipping Points are not committed upon crossing critical thresholds in global warming, as is often assumed. Instead, it is possible to temporarily overshoot a threshold without causing tipping, provided the duration of the overshoot is short. In this Idea, we demonstrate that restricting the time over $1.5^oC$ would considerably reduce tipping point risks.

The goal of the Paris Climate Agreement (2015) is to keep long-term global warming well below $2^oC$, and if possible, below $1.5^oC$, relative to pre-industrial levels. Global warming has already exceeded the $1.5^oC$ threshold for a period of 12 months (Copernicus Climate Change Service, 2024). Maintaining long-term warming at $1.5^oC$ necessitates decarbonisation rates that are highly unlikely given the current progress (Bossy et al., 2024). Therefore, at the very least, eventual stabilisation at this threshold suggests that a temporary overshoot of the $1.5^oC$ level is becoming increasingly likely.

Many elements of the climate system are vulnerable to large and abrupt changes, often referred to as tipping points (Lenton et al., 2008). Panel (a) of Figure 1 presents a probabilistic risk assessment for the number of elements of the climate system that could experience tipping at various levels of stabilised global warming. An uncertainty range is determined from the estimated ranges of each element's warming threshold location, as provided in a recent assessment of climate tipping points (Armstrong McKay et al., 2022), see Supplementary Material (Ritchie et al., 2025) for more details. Even if warming could be

stabilised at $1.5^oC$ without overshoot, the most likely scenarios indicate that between one and four elements may eventually tip (see Figure 1a). Stabilising warming at the upper limit of the Paris Climate Agreement (2015) ($2^oC$), raises the most probable range to between four and seven elements tipping. Current climate commitments are projected to result in global warming of $2.7 \pm 0.2^oC$ by the end of the century (Climate Action Tracker, 2024). Stabilising at this warming level could broaden the range to between seven and ten elements tipping. However, under these commitments, warming would continue beyond 2100,

and so if warming were only to stabilise near $4^oC$, the range shifts to between ten and thirteen major Earth system elements tipping (Figure 1a). Only the collapse of Arctic Winter sea ice and the East Antarctic ice sheet are deemed *very unlikely* to experience tipping.

However, system inertia means tipping is not committed once the tipping threshold is crossed, as is often implicitly assumed (Rockström et al., 2023). Instead, tipping can still be avoided if the exceedance of a threshold has a short duration compared to

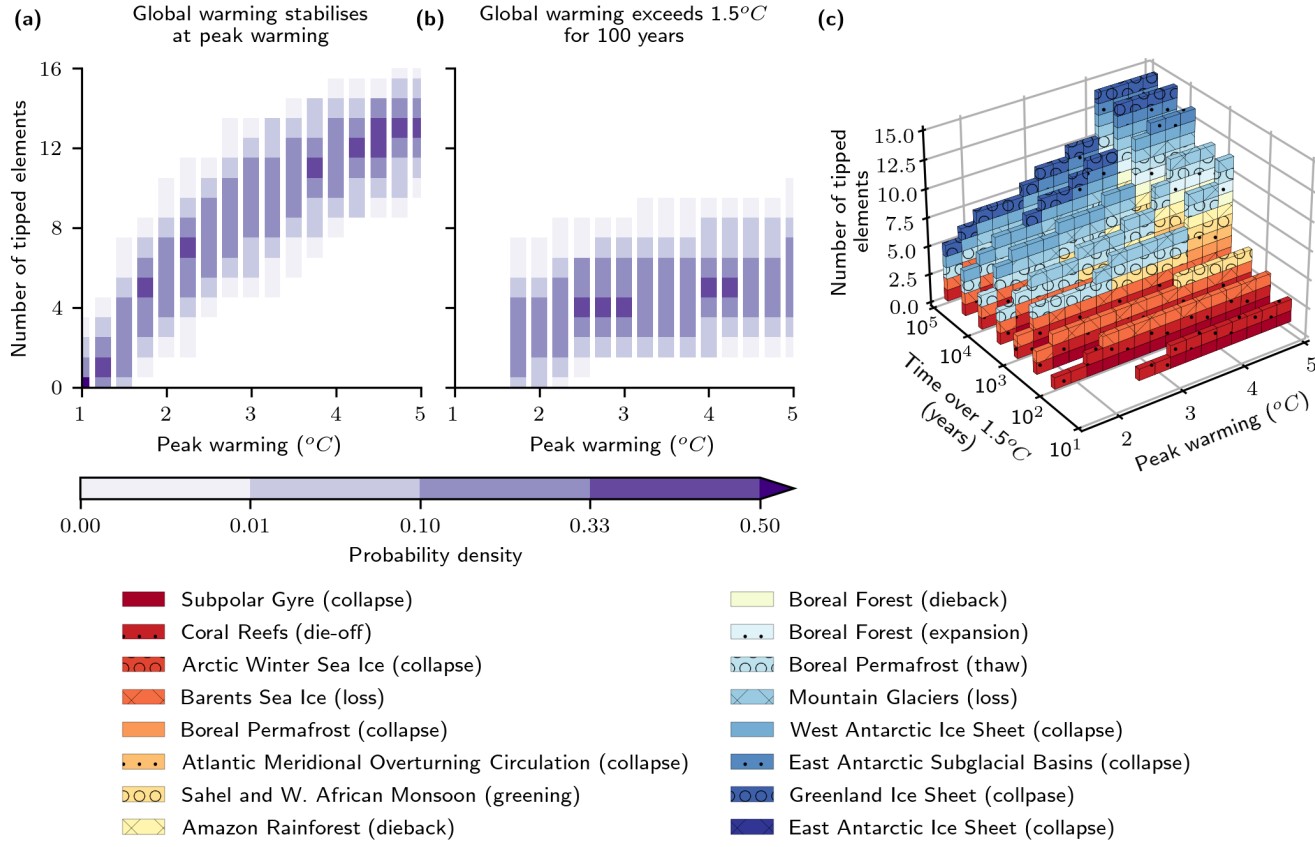

**Figure 1. Tipping risk of different climate system elements for different overshoot profiles.** Probabilistic number of elements of the climate system, binned by increments of $0.25^oC$, that undergo tipping for: (a) temperature stabilising; and (b) exceeding $1.5^oC$ for 100 years, for different levels of peak global warming. (c) 3-D projection of number of tipped elements based on the peak global warming and the time over $1.5^oC$ (note, on a logarithmic scale). Individual tipping elements are represented by colour and hatching with the colour coded according to their tipping timescale, with the fastest tipping elements in red and the slowest in blue. Uncertainty ranges given in (a) and (b) and represented by purple shading are calculated after fitting an exponentially modified Gaussian distribution to each of the tipping timescales and threshold values given in Armstrong McKay et al. (2022) by assuming that the lower, central, and upper estimates correspond to the 5%, 50% and 95% cumulative density levels respectively. Panel (c) uses the central estimates only to display the individual tipping elements.

the characteristic timescale of the tipping element. Previously, Ritchie et al. (2019) showed that tipping point risk depends on both the threshold temperature, represented by a fold bifurcation, and the timescale of the tipping element. In general, tipping elements with slow timescales allow overshoots that avoid tipping, whereas, fast tipping elements leave very little margin for overshoot without tipping (Ritchie et al., 2021). Given that the recent comprehensive study of Armstrong McKay et al. (2022) provides estimates of both timescales and critical warming thresholds for a broad set of Earth system components, we can now combine the analyses of Armstrong McKay et al. (2022) and Ritchie et al. (2021), to show the dependence of tipping point

risks on both peak global warming and also the duration of exceedance of $1.5^oC$. The Supplementary Material (Ritchie et al., 2025) details how the theory has been advanced to accommodate both the tipping timescale and exceedence of a predefined temperature (here $1.5^oC$) using a simple conceptual model, modified from Wunderling et al. (2021). Ideally, a third piece of information that details how quickly the stability of the initial stable state decreases as the tipping threshold approaches (for
instance, inferred from the curvature of the equilibrium branch) would be available. A faster decline in stability would mean that the tipping element is further from equilibrium and would help facilitate an overshoot that does not result in tipping. However, as this is not yet available, we have assumed that the parameter value is the same across all tipping elements. Future extensions of Armstrong McKay et al. (2022) may consider deriving the curvature parameter. The theory assumes a symmetric overshoot profile, however, as shown previously, more realistic, asymmetric profiles have also provided good agreement to the
theory (Ritchie et al., 2021).

    Panel (b) of Figure 1 is of identical format to panel (a), except that now warming only remains above $1.5^oC$ for 100 years. Comparing panel (b) to (a), shows a notable difference, with the number of elements that could undergo tipping considerably reduced. If global warming would peak at $2^oC$ but the time over $1.5^oC$ was limited to 100 years, the likely range of elements to tip drops to between two and four elements compared with four to seven if warming stabilised at $2^oC$. An overshoot of 100
45    years that reaches a peak warming of $3^oC$, would have the most likely result of four elements tipping, which compares to a range of eight to eleven if the warming instead stabilised at its peak.

    Panel (c) provides the 3-D picture of the individual elements that tip (now using the best estimates for thresholds and timescales given in Armstrong McKay et al. (2022)), for overshoot profiles characterised by both peak warming and time over $1.5^oC$ (panel animation provided as a supplement). Without the uncertainties, panels (a) and (b) would be cross sections
of panel (c), namely the left-hand "back wall" and the second visible row from the right-hand front, respectively. Panel (c) demonstrates that fast tipping elements (red blocks) would only avoid tipping if the overshoot duration is short (only decades over $1.5^oC$) combined with a small peak overshoot (notably the coral reefs and subpolar gyre are the most susceptible to tipping). For example, if the duration of exceedance of $1.5^oC$ is less than 30 years and the peak warming is less than $2.5^oC$ it may be possible to avoid all the tipping elements considered here (shown by the incomplete red bar at the front of panel (c)). In
contrast, the slowest tipping elements (blue blocks), are not committed to tip until the time over $1.5^oC$ approaches a thousand years, despite these elements possessing some of the lowest warming thresholds. Specifically, the Greenland ice sheet is found not to tip until the overshoot duration is greater than 10,000 years, which agrees well with simulations from two state-of-the-art numerical models (Bochow et al., 2023), despite assuming a simple conceptual model. The accumulation of tipped elements in the back corner of panel (c) is because this is where both peak warming and time over $1.5^oC$ are large. It is important to
highlight that not all of these overshoot profiles would be plausible, even with carbon dioxide removal technologies, particularly the front right corner. Specifically, considerations such as technical, economic and sustainability can limit the scales required at which carbon dioxide must be removed for such overshoots to be possible (Schleussner et al., 2024). Moving away from this front right corner and towards the back left corner coincides with increasing the feasibility of the overshoot profiles.

    In this study we do not consider the possibility that tipping elements can interact (Wunderling et al., 2024). However, some of
the tipping thresholds provided in the Armstrong McKay et al. (2022) study are likely to account for some of these interactions.

Furthermore, the threshold values may have been determined from transient climate simulations rather than corresponding to equilibrium values as assumed here, which would bias the results. We assume that the thresholds can be represented by a fold bifurcation, but this might not always be true which could lead to further uncertainty. Other factors to consider are the applicability of global warming as a forcing for all tipping elements and sensitivities to initial conditions (Mehling et al., 2024; Romanou et al., 2023).

Our study challenges the conventional wisdom that the commitment to tip occurs as soon as a critical threshold is crossed. Instead, the number of elements that would undergo tipping is severely reduced if the duration of exceedance of the Paris 1.5$^o$C can be kept below a century. Furthermore, this analysis suggests that all tipping elements could be avoided if global warming over 1.5$^o$C is restricted to 30 years and peak warming is kept below 2.5$^o$C.

*Video supplement.* Animation of Figure 1c provided as a supplement.

*Author contributions.* P.D.L.R. and C.H. designed and directed the Idea. P.D.L.R., C.H., and P.M.C, helped to shape the Idea and drafted the manuscript. P.D.L.R. performed the analysis and created the animation.

*Competing interests.* The authors declare no competing interests.

*Acknowledgements.* P.D.L.R. and P.M.C were supported by the Optimal High Resolution Earth System Models for Exploring Future Climate Changes (OptimESM) project, grant agreement number 101081193 and by ClimTip. This is ClimTip contribution #43; the ClimTip project has received funding from the European Union's Horizon Europe research and innovation programme under grant agreement No. 101137601: Funded by the European Union. Views and opinions expressed are however those of the authors only and do not necessarily reflect those of the European Union or the European Climate, Infrastructure and Environment Executive Agency (CINEA). Neither the European Union nor the granting authority can be held responsible for them. C.H. acknowledges the Natural Environment Research Council National Capability Fund awarded to the UK Centre for Ecology and Hydrology.

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
