# Peer review of "ESD Ideas: Climate tipping is not instantaneous – the duration of an overshoot matters"

_EGUsphere, 2024_

## Author Comment (AC1)

**ESD Ideas: Climate tipping is not instantaneous – the duration of an overshoot matters**
**Reviewer Responses**

We are grateful for the constructive reviewer comments received on our manuscript. These comments are repeated in black, and our responses are given in blue.

**Response to Reviewer 1**

Ritchie et al. use a conceptual/mathematical framework to quantify which tipping elements would tip when overshooting 1.5$^o$C global warming on different timescales. The article is concise and well-written, and the methods are clear and well-established. The topic of climate overshoots is obviously very relevant and well-suited for ESD. However, I have some concerns regarding the framing of the results, especially given the expected broad readership, and the lack of an assessment of uncertainties, which should however be straightforward to implement.

**General/major comments**

**1. Novelty:** This may of course be due to the length constraints, but the idea does not strike me as particularly "innovative" per the description of the "ESD Ideas" format. For example, overshoots of the Paris Agreement have previously been explored in a more comprehensive setup (including idealized interactions between different tipping elements) by Wunderling et al. (2023), which is not cited. In my view, this does not preclude publication of the present manuscript, but the authors should motivate the need for this piece more explicitly and position it better within the existing literature. One option would be to target a broader/policy-oriented audience. In this case, the following point would be especially relevant.

Yes, we intend to target a broader and policy-oriented audience to highlight the importance that tipping is not committed upon crossing the threshold. The novelty of the Idea is that for the first time it provides an assessment for the implications of overshoot for all tipping elements (as opposed to a select few as previously considered) identified in the Armstrong-McKay et al. (2022) study. Interactions are not included in this study and that is now mentioned in the manuscript citing the more recent review by Wunderling et al. (2024), however, the thresholds are at least partly informed by climate model simulations which have some interactions built in.

**2. Policy message:** Since this is a short article which I expect to be accessible to a rather wide audience, it would be important to both consider and explicitly acknowledge its policy implications and applicability.

Most importantly, the article should be very clear upfront that some (many?) of the underlying overshoot trajectories are neither physically nor socio-economically plausible. For example, a 4–5$^o$C overshoot before a 1.5$^o$C global warming stabilization within 100 years seems clearly unfeasible, but the avoidance of tipping under such unrealistic scenarios could a give a false sense of security that strong warming in the medium term would not be as dangerous. The recent paper by Schleussner et al. (2024) is worth a read (and maybe a cite) in this regard. They conclude: "[T]echnical, economic and sustainability considerations may limit the realization of carbon dioxide removal deployment at such scales [of several hundred gigatonnes]. Therefore, we cannot be confident that temperature decline after overshoot is achievable within the timescales expected today."

To prevent any misinterpretation by readers or decision-makers, I think it is essential that this article cautions which of the results might be applicable to current global warming and which are not.

The policy message of the manuscript has been enhanced by providing a new opening paragraph to put the study into context. We also agree with the reviewer that some scenarios are unrealistic (without large scale geoengineering intervention) and so we now write "It is important to highlight that not all of these overshoot profiles would be plausible, even with carbon dioxide removal technologies, particularly the front right corner. Specifically, considerations such as technical, economic and sustainability can limit the scales

required at which carbon dioxide must be removed for such overshoots to be possible (Schleussner et al., 2024). Moving away from this front right corner and towards the back left corner coincides with increasing the feasibility of the overshoot profiles." Note that identifying the precise boundary for when overshoot scenarios become unfeasible requires further research.

**3. Minimal assessment of uncertainties:** Overshoot timescales of the order of a human lifespan (Fig. 1b) are probably most interesting and relevant for a broad readership. However, assessing overshoots on these timescales in a non-probabilistic way could give a distorted view of the associated risks. In addition, the article is currently not clear about this omission: for example, "three are likely to tip" in L41-42 suggests that some probabilistic assessment might have taken place.

Since Armstrong McKay et al. (2022) provide confidence intervals for all GW level estimates and for most timescale estimates, it should be straightforward to turn the deterministic Fig. 1b into a probabilistic one, which could show a probability distribution (e.g., as a violin plot) of the number of tipped elements for each global warming level when taking into account parameter uncertainty. This would mean that it could not be shown which elements would tip in each case, but I would deem that an acceptable trade-off since the current Fig. 1b is already a part of Fig. 1c. I think it is ok to keep Fig. 1c as it is, but to remind the reader in the text that uncertainties are neglected here.

In Fig 1a and 1b we now provide an assessment of uncertainties, and have kept Fig 1c the same as suggested. Instead of violin plots, we have used colour to represent the probability density in an effort to keep the figure as simple as possible. Detailing the methods, we now write at the end of the Supplementary Material "Panels (a) and (b) in Figure 1 of the main paper, plot the probability density for the number of tipped elements in colour. An exponentially modified Gaussian distribution is fitted, using least squares, to each element's tipping timescale and threshold location. The lower, best, and upper estimates are assumed to correspond to the 5, 50, and 95% cumulative probability density levels. In the scenario where no estimate is given for either the upper or lower value, the best estimate is used instead. A random sample is then drawn from each of the 32 distributions to provide the threshold location and tipping timescale for each of the 16 tipping elements. Assuming these threshold and timescale values the inverse square law is used as previously to calculate the number of elements that would tip for each warming scenario. The process is repeated 1,000 times to thus generate the probability density for the number of tipped elements for each scenario."

The results remain qualitatively the same; importantly, there is still a large reduction in the number of elements that would undergo tipping if the overshoot duration is restricted to 100 years (Fig 1b) compared with stabilising the temperature at its peak (Fig 1a). The main text and figure caption has been re-worked to reflect these changes.

**Specific and minor comments**

L2–3: "as is often assumed": please provide one or two references that this is indeed the case – maybe from the scientific literature, or from media or the policy sphere. This is an important point because the entire article addresses an alleged common misconception that tipping would be instantaneous, but it is not shown that this misconception actually exists.

We have now added a reference in the main text where a similar statement appears as opposed to in the abstract. We have made a slight reformulation to mention that a commitment to tipping is often assumed once a threshold has been crossed. Some other examples that make this assumption include "A tipping point is when a temperature threshold is passed, leading to unstoppable change in a climate system..." (World on brink of five 'disastrous' climate tipping points, Guardian Sept 2022); "Crossing Earth system tipping points would have "catastrophic" impacts on societies" (Q&A: Climate tipping points have put Earth on 'disastrous trajectory', Carbon Brief Dec 2023); "Crossing these thresholds would disrupt the Earth's systems triggering the collapse of ice sheets" (Climate change: Six tipping points 'likely' to be crossed, BBC News Sept 2022). Note in the literature and media, a tipping point is often referred to as the threshold.

L16: The number calculated by the Climate Action Tracker refers to 2100 and the websites explicitly states "Temperatures continue to rise after 2100" (https://climateactiontracker.org/publications/the-climate-crisis-worsens-the-warming-outlook-stagnates/). Since you are discussing stabilization values in this paragraph, could you contextualize this?

This is a good point and we now use this to motivate why we also consider stabilising at $4^oC$ warming, by writing "Current climate commitments have been estimated to lead to global warming reaching $2.7 \pm 0.2^oC$ by the end of the century (Climate Action Tracker, 2024). Stabilising at this level of warming would increase the range to between seven and ten elements tipping, but under these commitments, warming would continue beyond 2100 and so if temperature was to only stabilise near $4^oC$, then the range is between ten and thirteen ..."

Fig. 1: Some tipping elements are hard to distinguish just by color, maybe use colors in combination with different hatching patterns?

We have followed your suggestion and in addition use four different hatching patterns to distinguish more clearly between the different tipping elements.

Fig. 1c: I am unsure if the current choice of the logarithmic axis is ideal. An overshoot of $10^5$ years may at most be relevant for paleoclimate if we frame interglacials as "overshoots" to a glacial background climate, but probably not for the current anthropogenic global warming. This does not mean that the figure needs to be extensively modified, but the large range of overshoot durations should be pointed out (and put in context) explicitly in the text and/or caption. And maybe consider cutting off the axis at (some) 1000 years?

The "back wall" of Fig. 1c is designed to effectively consider the scenario of no overshoot (i.e. it corresponds to the cross section of panel (a) without uncertainties). We explicitly make reference to this by writing in the main text "Without the uncertainties panels (a) and (b) would be cross sections of panel (c), namely the left-hand "back wall" and the second visible row from the right-hand front, respectively." Furthermore, we want to highlight that the slow tipping elements only undergo tipping for very long overshoots and even cutting the axis at $10^4$ years the Greenland ice sheet would not be visible under any scenario.

L26 and following: Are the values in Armstrong McKay et al. really derived from an equilibrated climate (as suggested by Fig. 1a) or from transient simulations? If they are derived from transient runs, wouldn't this lead to a systematic bias when they are used in an equilibrium view?

This is a good point. It is not clear in the Armstrong-McKay et al. (2022) study if the thresholds are derived from an equilibrated climate or transient simulations. However, some of the systems with very long timescales (i.e. major ice sheets) have low thresholds, which would indicate that they are derived from an equilibrated climate. Nevertheless, we do now write in the main text "Furthermore, the threshold values may have been determined from transient climate simulations rather than corresponding to equilibrium values as assumed here, which would bias the results." to acknowledge this needs to be considered as a possibility.

L29: It would be more reader-friendly to include the Supplement directly within the article (e.g., as an Appendix), if this is allowed by the Editors.

We understand this is not possible for ESD Ideas, but can if allowed.

L30 and Suppl. L38: It is not entirely clear how the choice $\alpha = 1$ for all TEs is motivated. Could you check the validity of this assumption, for example, in one or two commonly used conceptual models of some tipping elements (e.g., Stommel model for the AMOC)?

The choice of $\alpha = 1$ is arbitrary and we suggest that this should be an area of future research as the information is not currently available in the Armstrong-McKay et al. (2022) study. Note in the previous version a small mistake was found that meant the $\alpha$ would not affect the overshoot duration (as the scaling on the

forcing cancelled out the change of curvature). We have therefore reworked the model slightly, including introducing two parameters. A scaling parameter, $\beta$, that does not feature in the inverse square law and the $\alpha$ that still provides a measure of the curvature of the equilibrium curve. We still set $\alpha = 1$ and so the results are unchanged. We have also checked how the allowed overshoot duration compares with that which can be calculated via the Stommel-Cessi AMOC model (Cessi, 1994). If the best estimates of the Armstrong-McKay et al. (2022) study are used then our choice of $\alpha = 1$ corresponds to an order of magnitude stricter condition on the square of the overshoot duration than the Stommel-Cessi model would suggest (a tipping timescale of 100 years would make it the same order of magnitude, but still more restrictive). In the Supplementary we now write "Note that this choice of $\alpha$ corresponds to a conservative choice for the AMOC according to the Stommel-Cessi model (Cessi, 1994), which would allow the square of the overshoot duration to be an order of magnitude larger. However, further research is required to determine the curvature parameter for all tipping elements."

Recently, some overshoot studies have been performed with state-of-the-art numerical models, notably Bochow et al. (2023) for the Greenland ice sheet (see their Fig. 3 for different overshoot durations). It would be very valuable to benchmark (e.g., in the Supplement) your estimates from Fig. 1c against their results to see if the results hold when compared to a comprehensive model.

The results in Fig 1c compare well with the Bochow et al. (2023) study. There it is shown that the Greenland ice sheet does not tip until peak summer temperatures above present day exceed 4 and $5^o$C in PISM and YELMO respectively for a 10,000 year convergence time. We think this is an important validation of the theory and so we now write in the main text "Specifically, the Greenland ice sheet does not tip until the overshoot duration is greater than 10,000 years, which agrees well with simulations from two state-of-the-art numerical models (Bochow et al., 2023)."

"Climate Action Tracker" reference should be updated to the most recent version (and equipped with a URL).

We have updated the Climate Action Tracker reference to 2024 and included a URL.

Suppl. L23: "as expecting" $\rightarrow$ "as expected"

Thank you for spotting this, it has now been corrected.

Bochow, N., Poltronieri, A., Robinson, A., Montoya, M., Rypdal, M., & Boers, N. (2023). Overshooting the critical threshold for the Greenland ice sheet. Nature, 622, 528–536. https://doi.org/10.1038/s41586-023-06503-9

Schleussner, C.-F., Ganti, G., Lejeune, Q., Zhu, B., Pfleiderer, P., Prütz, R., et al. (2024). Overconfidence in climate overshoot. Nature, 634, 366–373. https://doi.org/10.1038/s41586-024-08020-9

Wunderling, N., Winkelmann, R., Rockström, J., Loriani, S., Armstrong McKay, D. I., Ritchie, P. D. L., et al. (2023). Global warming overshoots increase risks of climate tipping cascades in a network model. Nature Climate Change, 13, 75–82. https://doi.org/10.1038/s41558-022-01545-9

---

## Author Comment (AC2)

**ESD Ideas: Climate tipping is not instantaneous – the duration of an overshoot matters**
**Reviewer Responses**

We are grateful for the constructive reviewer comments received on our manuscript. These comments are repeated in black, and our responses are given in blue.

**Response to Reviewer 2**

This Idea makes the relevant point that climate tipping elements do not tip instantly upon crossing a critical threshold in global mean temperature – the tipping risk depends on the duration of the overshoot as well as the peak exceedance value. The authors illustrate this statement by calculating whether various proposed tipping elements tip for different combinations of exceedance duration and peak warming, based on a simplified formula (Ritchie et al. 2019) and threshold/timescale estimates of Armstrong McKay et al. (2022).

The text is clearly written, and the results coherently support the main argument. I agree with the authors that it is crucial to consider the timescales of the forcing relative to the internal timescales of the tipping element for assessing tipping risk, an aspect sometimes ignored in the broader debate. However, I am concerned that the article oversimplifies the problem of climate tipping under overshoots and is therefore misleading particularly for a non-specialist audience. Without additional contextualization, there is a risk that the article perpetuates misunderstandings about tipping points rather than correcting them. I believe this can be solved by replacing some of the descriptive text with a brief, yet critical discussion of the assumptions, uncertainties and real-world complexities involved.

**Major comment**

My main critique is that the results described in this work convey a deceptive sense of certainty. The presentation suggests that whether a tipping element will tip can be determined with certainty from knowledge of the peak warming level and duration above 1.5 degrees C, independently of the other tipping elements. This depiction neglects that

- The warming threshold and timescale estimates in Armstrong McKay et al. (2022) have considerable uncertainty ranges.

- A sharp critical threshold may not exist for all tipping elements.

- Besides peak and duration, the shape of the forcing protocol may matter.

- For a given forcing protocol, tipping may be sensitive to the initial condition (see e.g. Romanou et al. 2023, Mehling et al. 2024).

- The climate tipping elements interact (Wunderling et al. 2024). The fact that interactions are neglected here is only mentioned in the Supplementary Material.

- Global warming levels are a climate response to forcing themselves and may not be an appropriate control parameter for all tipping elements considered.

This implies that the results in Fig. 1 have a significant uncertainty themselves and should be viewed probabilistically. While I think the results are still interesting in their current form, I ask the authors to better clarify in the main text that the results are simply the outcome of combining the best estimates of Armstrong McKay et al. (2022) with the adapted inverse-square law under multiple simplifying assumptions, and to highlight the inherent uncertainties.

In Fig 1a and 1b we now provide an assessment of uncertainties, but we have kept Fig 1c as previously so that the individual tipping elements can still be identified. We have used colour to represent the probability density for the number of elements tipped for each scenario. Detailing the methods, we now write at the end of the Supplementary Material "Panels (a) and (b) in Figure 1 of the main paper, plot the probability

density for the number of tipped elements in colour. An exponentially modified Gaussian distribution is fitted, using least squares, to each element's tipping timescale and threshold location. The lower, best, and upper estimates are assumed to correspond to the 5, 50, and 95% cumulative probability density levels. In the scenario where no estimate is given for either the upper or lower value, the best estimate is used instead. A random sample is then drawn from each of the 32 distributions to provide the threshold location and tipping timescale for each of the 16 tipping elements. Assuming these threshold and timescale values the inverse square law is used as previously to calculate the number of elements that would tip for each warming scenario. The process is repeated 1,000 times to thus generate the probability density for the number of tipped elements for each scenario."

The results remain qualitatively the same; importantly there is still a large reduction in the number of elements that would undergo tipping if the overshoot duration is restricted to 100 years (Fig 1b) compared with stabilising the temperature at its peak (Fig 1a). The main text and figure caption has been re-worked to reflect these changes.

We have commented that "The theory assumes a symmetric overshoot profile, however, as shown previously, more realistic, asymmetric profiles have also provided good agreement to the theory (Ritchie et al., 2021)". Although we do not consider interactions directly the thresholds themselves are at least partly informed by climate model simulations which have some interactions built in. Nevertheless, we still note that we do not directly consider interactions as well as your other three remaining points that will lead to further uncertainties by writing in the penultimate paragraph, "In this study we do not consider the possibility that tipping elements can interact (Wunderling et al., 2024). However, some of the tipping thresholds provided in the Armstrong McKay et al. (2022) study are likely to account for some of these interactions. Furthermore, the threshold values may have been determined from transient climate simulations rather than corresponding to equilibrium values as assumed here, which would bias the results. We assume that the thresholds can be represented by a fold bifurcation, but this might not always be true which could lead to further uncertainty. Other factors to consider are the applicability of global warming as a forcing for all tipping elements and sensitivities to initial conditions (Mehling et al., 2024; Romanou et al., 2023)."

**Specific comments**

L. 53: The authors frame their Idea as a challenge of the "conventional wisdom" that tipping occurs instantaneously. However, the authors' previous work (Ritchie et al. 2019, 2021) and related studies (Bochow et al. 2023, Wunderling et al. 2023) have already debunked this "wisdom" and established the idea in the field over the past years. This questions how innovative the idea is in this work. If the authors' aim is thus to establish the concept among a wider audience, clarifying the underlying assumptions for obtaining Fig. 1 is especially important.

We have made a slight reformulation that the "conventional wisdom" is that tipping is committed once the threshold is crossed. We have added a high profile reference to support this statement at the first opportunity in the main text. The novelty of the Idea is that for the first time it provides an assessment for the implications of overshoot for all tipping elements (as opposed to a select few as previously considered) identified in the Armstrong-McKay et al. (2022) study. However, we do indeed intend to target a broader and policy-oriented audience to establish this concept more widely.

Could the authors comment on how their results might depend on the specific shape of the forcing protocol, i.e. steepness, asymmetry, etc.?

Although the theory assumes a symmetric forcing profile, the theory has previously been shown to align well for more realistic overshoot profiles that are asymmetric. We now write "The theory assumes a symmetric overshoot profile, however, as shown previously asymmetric profiles have also provided good agreement to the theory (Ritchie et al., 2021)".

L. 17-19: In my opinion, counting the number of Earth system elements tipping is not very informative, as

the tipping elements have highly heterogeneous impacts. Perhaps the cumulative impact of multiple elements tipping could be viewed in a different way?

This is a good suggestion, but would require further research. Furthermore, we want to highlight the importance of tipping timescales, and identifying individual tipping elements helps emphasise that the fast tipping elements are those that are at greater risk of tipping.

The reference list seems rather selective, perhaps owing to the short format of the Idea. I note that all scientific articles cited in the main text are led by the same institution. Additional references underlying the authors' approach (e.g., Wunderling et al. 2021) are only cited in the Supplementary Material. If allowed by the format, adding further references to the main text would help putting this article into context.

We now provide a greater variety of references, but the Ideas format is limited to 15 references.

**Technical comments**

L. 16: Ref. Climate Action Tracker (2021) – is this a reference from the peer-reviewed literature and, if not, could the specific study be cited here?

We have updated the reference to 2024 and also now provide a URL to the resource.

L. 1: The figure labels are almost too small to read.

The figure labels have been increased to match the size of the text in the manuscript.

L. 25: . . . without tipping (Ritchie et al. 2021).

This has been corrected as suggested.

L. 28: . . . of 1.5C. Our. . .

Punctuation has now been added.

L. 31-32: The relevance of the curvature of the equilibrium branch for tipping risk is not explained. Only in the Supplementary Material it is stated that a fold bifurcation is assumed for all tipping elements. This assumption may not hold for all tipping elements catalogued in Armstrong McKay et al. (2022).

The curvature of the equilibrium branch informs how quickly the stability of the equilibrium is changing. A faster decrease would cause a greater lag (system would be further from equilibrium) and so would help facilitate an overshoot without tipping. We now write "Ideally, a third piece of information that details how quickly the stability of the initial stable state decreases as the tipping threshold approaches (for instance, inferred from the curvature of the equilibrium branch) would be available. A faster decline in stability would mean that the tipping element is further from equilibrium and would help facilitate an overshoot that does not result in tipping." We also now note on multiple occasions that the threshold is assumed to be represented by a fold. We write that this could lead to further uncertainty "We assume that the thresholds can be represented by a fold bifurcation, but this might not always be true which could lead to further uncertainty."

L. 39: No comma after Barents sea ice

This paragraph has been heavily revised after the inclusion of an uncertainty assessment and so this sentence no longer exists.

L. 44: Period missing after supplement)

Punctuation has now been added.

L. 46-47: The sentence "These fast tipping elements would..." seems redundant, especially given the final sentence (L. 55-56), and the space could be used to comment on limitations/uncertainties instead.

Agreed, we have deleted the sentence as suggested.

Supplementary Material

L. 3: "...elements of the Earth system, and based..." – should the "and" be there?

Thank you the "and" has now been removed.

L. 23: change "expecting" to "expected"

This has been corrected.

Bochow et al., "Overshooting the critical threshold for the Greenland ice sheet" (2023) https://doi.org/10.1038/s41586-023-06503-9

Mehling et al., "Limits to predictability of the asymptotic state of the Atlantic Meridional Overturning Circulation in a conceptual climate model" (2024) https://doi.org/10.1016/j.physd.2023.134043

Romanou et al., "Stochastic Bifurcation of the North Atlantic Circulation under a Midrange Future Climate Scenario with the NASA-GISS ModelE " (2023) https://doi.org/10.1175/JCLI-D-22-0536.1

Wunderling et al., "Global warming overshoots increase risks of climate tipping cascades in a network model" (2023) https://doi.org/10.1038/s41558-022-01545-9

Wunderling et al., "Climate tipping point interactions and cascades: a review" (2024) https://doi.org/10.5194/esd-15-41-2024

---

## Author Comment (AC3)

**ESD Ideas: Climate tipping is not instantaneous – the duration of an overshoot matters**
**Reviewer Responses**

We are grateful for the constructive reviewer comments received on our manuscript. These comments are repeated in black, and our responses are given in blue.

**Response to Reviewer 3**

Ritchie et al. describe the concept of overshooting tipping points in their ESD ideas. They state that crossing a tipping point is not an instantaneous event but rather emerges over a finite time. This allows, in theory, for the prevention of a tipping point by reversing the forcing, i.e., global warming, to previous levels. They demonstrate that restricting the time above 1.5°C above pre-industrial levels would reduce risks associated with tipping points. For this, they combine results from a recent review paper about tipping elements (McKay et al., 2022) and from an earlier conceptual study about the overshooting concept by Ritchie et al. (2021).

The manuscript is clearly written and understandable for a broader audience. I fully agree that the topic of overshooting tipping points and considering their timescales is generally important. I also agree with the authors that the notion of the instantaneousness of tipping points is no longer appropriate. At the same time, I struggle to see the novelty of the presented concepts/results, at least for the scientific community. While I see that this misconception of the instantaneousness of tipping points is very prevalent in the public media/opinion, I am not sure if this is also the case in recent scientific publications; at least some references would be needed to back up this claim. Alternatively, a reframing of the presented work in relation to the misconception in the public rather than the scientific community might need to be considered.

Additionally, I think the manuscript partly oversimplifies the tipping point/overshoot concept in a manner that could convey overconfidence in the results, especially for non-experts. This can be easily avoided by stating the limitations of the results and discussing uncertainties.

**General comments**

I struggle to see the novelty of the presented concept, as I do not agree with the authors that this is very new information. As I understand it, the ESD Ideas format is intended for novel ideas or concepts. However, the idea that tipping is not instantaneous is not new (e.g., their own paper Ritchie et al., 2021), and it is not surprising to me that tipping can be avoided in conceptual models when the overshoot temperature and time are limited. I could imagine this manuscript, including the supplement, being included as a section in a more detailed manuscript.

The novelty of the Idea is that for the first time it provides an assessment for the implications of overshoot for all tipping elements (as opposed to a select few as previously considered) identified in the Armstrong-McKay et al. (2022) study. We intend to target a broader and policy-oriented audience to highlight the importance that tipping is not committed upon crossing the threshold. Therefore we have written a new opening paragraph to contextualise the study more clearly.

It seems like uncertainties related to the threshold levels are not mentioned anywhere in the main text. I think it should at least be stated that there is considerable uncertainty related to tipping point thresholds, even in McKay et al. (2022). Generally, I feel that the manuscript is written in a way that takes the concept of tipping elements/points for granted without considering the uncertainties. This gives the impression that the results shown are considered to be absolute truth. I wonder if it is possible to repeat the analysis with upper and lower limits for the individual tipping point thresholds and show them in the supplementary information or in an extended figure.

In Fig 1a and 1b we now provide an assessment of uncertainties, but keep Fig 1c the same. Detailing the methods, we now write at the end of the Supplementary Material "Panels (a) and (b) in Figure 1 of the

main paper, plot the probability density for the number of tipped elements in colour. An exponentially modified Gaussian distribution is fitted, using least squares, to each element's tipping timescale and threshold location. The lower, best, and upper estimates are assumed to correspond to the 5, 50, and 95% cumulative probability density levels. In the scenario where no estimate is given for either the upper or lower value, the best estimate is used instead. A random sample is then drawn from each of the 32 distributions to provide the threshold location and tipping timescale for each of the 16 tipping elements. Assuming these threshold and timescale values the inverse square law is used as previously to calculate the number of elements that would tip for each warming scenario. The process is repeated 1,000 times to thus generate the probability density for the number of tipped elements for each scenario."

The results remain qualitatively the same; importantly there is still a large reduction in the number of elements that would undergo tipping if the overshoot duration is restricted to 100 years (Fig 1b) compared with stabilising the temperature at its peak (Fig 1a). The main text and figure caption has been re-worked to reflect these changes.

I do not see any mention in the main text that the results are based on a, arguably, very simple conceptual model.

We now mention multiple times that the results are based on a simple conceptual model. However, we also note that the results align well with simulations from two state-of-the-art models for the Greenland ice sheet despite using a simple conceptual model. We write, "Specifically, the Greenland ice sheet is found not to tip until the overshoot duration is greater than 10,000 years, which agrees well with simulations from two state-of-the-art numerical models (Bochow et al., 2023), despite assuming a simple conceptual model."

Generally, I think some more references are needed for some statements (see also specific comments).

We have now added some more references including a reference to support the statement about the assumptions that tipping is committed upon crossing the threshold.

I don't think the supplementary information is easy enough to access/find. Additionally, the supplement is almost as long as the ESD Ideas itself. I wonder why this is not expanded upon and submitted as a regular research article.

The Supplementary Material is provided in the list of references including a URL, which takes the reader directly to the PDF. The Idea at its core is to highlight, to a wider audience, the importance of accounting for timescales when considering overshoot scenarios by providing a comprehensive assessment for all tipping elements identified in the Armstrong-McKay et al. (2022) study. We think this makes the study align well with the format of ESD Ideas.

Coupling between the elements is ignored, but it is only stated in the supplement. It should at least be stated in the main text. Although, I wonder how much value the results have if coupling between the tipping elements is completely ignored, given that we know there is interaction between the individual elements (e.g. Wunderling et al., 2024).

Interactions are not included in this study and that is now mentioned in the main manuscript, citing Wunderling et al. (2024). However, the thresholds are, at least partly, informed by climate model simulations which have some interactions built in.

Given the broad target audience and policy relevance of the manuscript, I think it would be a good idea to comment on the feasibility of the presented overshoot scenarios. A recent study showed that there likely is an overconfidence in overshoot scenarios and temperature decline after an overshoot might not even be possible (Schleussner et al. 2024).

We agree with the reviewer, and have now added the following to address the feasibility of the overshoot

scenarios, "It is important to highlight that not all of these overshoot profiles would be plausible, even with carbon dioxide removal technologies, particularly the front right corner. Specifically, considerations such as technical, economic and sustainability can limit the scales required at which carbon dioxide must be removed for such overshoots to be possible (Schleussner et al., 2024). Moving away from this front right corner and towards the back left corner coincides with increasing the feasibility of the overshoot profiles.".

**Specific comments**

L.1: "as is often assumed" — I would expect some references for that claim. In fact, there have been recent studies showing that tipping is generally not instantaneous (e.g., Bochow et al. (2023) or Höning et al. (2024)). I would almost claim that within the broad scientific community, it is known that tipping is not instantaneous.

We have reformulated slightly to mention that a commitment to tipping is often assumed once a threshold has been crossed. We have now added a reference in the main text where a similar statement appears as opposed to in the abstract. Some other examples that make similar assumptions include "A tipping point is when a temperature threshold is passed, leading to unstoppable change in a climate system..." (World on brink of five 'disastrous' climate tipping points, Guardian Sept 2022); "Crossing Earth system tipping points would have "catastrophic" impacts on societies" (Q&A: Climate tipping points have put Earth on 'disastrous trajectory', Carbon Brief Dec 2023); "Crossing these thresholds would disrupt the Earth's systems triggering the collapse of ice sheets" (Climate change: Six tipping points 'likely' to be crossed, BBC News Sept 2022). Note in the literature and media, a tipping point is often referred to as the threshold.

L.8: There is no reference for the Barents Sea ice threshold. Is this also taken from McKay et al. (2022)?

Yes it was, but now due to the inclusion of an uncertainty assessment this whole paragraph has been reworked.

L.47: I am not sure if defining "extremely short" as several decades is appropriate.

We have removed "extremely".

L.53: Here again, I would expect references for the statement: "conventional wisdom that the commitment to tip occurs as soon as a critical threshold is crossed." I don't agree that this is "conventional wisdom" anymore.

We have provided a reference earlier when this statement is first given in the main text.

L.38 (supplement): How realistic is this choice of $\alpha$, and what influence does it have on the results? I would guess that for some tipping elements, e.g., Antarctica (Garbe et al. (2020)) or Greenland (Bochow et al. (2023), Höning et al. (2023 and 2024)), the curvature parameter could be roughly estimated.

The choice of $\alpha = 1$ is arbitrary and we suggest that this should be an area of future research as the information is not currently available in the Armstrong-McKay et al. (2022) study. Note in the previous version a small mistake was found that meant the $\alpha$ would not affect the overshoot duration (as the scaling on the forcing cancelled out the change of curvature). We have therefore reworked the model slightly, including introducing two parameters. A scaling parameter, $\beta$, that does not feature in the inverse square law and the $\alpha$ that still provides a measure of the curvature of the equilibrium curve. We still set $\alpha = 1$ and so the results are unchanged. We have also checked how the allowed overshoot duration compares for to that which can be calculated via the Stommel-Cessi AMOC model (Cessi, 1994). If the best estimates of the Armstrong-McKay et al. (2022) study are used then our choice of $\alpha = 1$ corresponds to an order of magnitude stricter condition on the square of the overshoot duration than the Stommel-Cessi model would suggest (a tipping timescale of 100 years would make it the same order of magnitude, but still more restrictive). In the Supplementary we now write "Note that this choice of $\alpha$ corresponds to a conservative choice for the AMOC according to the Stommel-Cessi model (Cessi, 1994), which would allow the square of the overshoot duration to be an order of magnitude larger. However, further research is required to determine the curvature parameter for

all tipping elements."

The colors in the plot are a little bit hard to distinguish. I wonder if a different color palette would make the plot more accessible.

We have now added four different hatching styles to make it easier to distinguish between the different tipping elements.

Garbe, J., Albrecht, T., Levermann, A. et al. The hysteresis of the Antarctic Ice Sheet. Nature 585, 538–544 (2020). https://doi.org/10.1038/s41586-020-2727-5

Dennis Höning et al 2024 Environ. Res. Lett. 19 024038, 10.1088/1748-9326/ad2129

Höning, D., Willeit, M., Calov, R., Klemann, V., Bagge, M., & Ganopolski, A. (2023). Multistability and transient response of the Greenland ice sheet to anthropogenic CO2 emissions. Geophysical Research Letters, 50, e2022GL101827. https://doi.org/10.1029/2022GL101827

Bochow, N., Poltronieri, A., Robinson, A. et al. Overshooting the critical threshold for the Greenland ice sheet. Nature 622, 528–536 (2023). https://doi.org/10.1038/s41586-023-06503-9

Ritchie, P.D.L., Clarke, J.J., Cox, P.M. et al. Overshooting tipping point thresholds in a changing climate. Nature 592, 517–523 (2021). https://doi.org/10.1038/s41586-021-03263-2

David I. Armstrong McKay et al., Exceeding 1.5°C global warming could trigger multiple climate tipping points.Science377, eabn7950(2022).DOI:10.1126/science.abn7950

Schleussner, CF., Ganti, G., Lejeune, Q. et al. Overconfidence in climate overshoot. Nature 634, 366–373 (2024). https://doi.org/10.1038/s41586-024-08020-9

Wunderling, N., von der Heydt, A. S., Aksenov, Y., Barker, S., Bastiaansen, R., Brovkin, V., Brunetti, M., Couplet, V., Kleinen, T., Lear, C. H., Lohmann, J., Roman-Cuesta, R. M., Sinet, S., Swingedouw, D., Winkelmann, R., Anand, P., Barichivich, J., Bathiany, S., Baudena, M., . . . Willeit, M. (2024). Climate tipping point interactions and cascades: A review. Earth System Dynamics, 15(1), 41–74. https://doi.org/10.5194/esd-15-41-2024

---

## Author Response (AR2)

**ESD Ideas: Climate tipping is not instantaneous – the duration of an overshoot matters**
**Reviewer Responses**

**Editor**

Dear authors,

The reviewers were quite satisfied with your revised manuscript, and there are only a few minor suggestions for improvement. I therefore decided for publish subject to minor revisions that are reviewed by me.

Looking forward to the published article.

Best,
Axel

Dear Editor,

We are pleased to hear that the reviewers were satisfied with the last round of revisions and that you have decided to publish subject to minor revisions. We have now carefully addressed the remaining comments, including those regarding the curvature parameter in our model. This has helped us to establish that our model was over parameterised and thus, without loss of generality, we can set $\alpha = 1$, as changing $\alpha$ would in effect be changing the tipping timescale.

We look forward to seeing our manuscript published in Earth System Dynamics soon.

Yours sincerely,

Paul Ritchie (on behalf of all authors)

**Response to Reviewer 1**

We are grateful for the further constructive reviewer comments received on our manuscript. These comments are repeated in black, and our responses are given in blue.

I would like to thank the authors for carefully addressing the reviewers' concerns, and the new version of the manuscript is much improved, especially regarding the many caveats of the simple conceptual modelling approach taken here. The uncertainty analysis in figures 1a+b has also improved the robustness of the results.

I have a few remaining concerns specifically regarding the curvature parameter:

- From my understanding, the curvature parameter alpha turns out to be very important for the overshoot duration, as the overshoot timescale scales with 1/alpha (equation 7 of the Supplementary Material). While the curvature parameter is mentioned in the manuscript, the discussion in L33-38 of the revised manuscript is unnecessarily vague. It would be best if this scaling was explicitly acknowledged in the main text and acknowledged as an important uncertainty for the estimate of safe overshoot durations.

- While I appreciate that the authors have looked into the issue of setting alpha=1 in a conceptual AMOC model, it would be useful if this example was actually worked out in a bit more detail in the supplementary material ("one order of magnitude larger" is a bit too vague, and it is unclear how alpha was derived in this conceptual model).

- In addition, I wonder if (a) the Stommel model is a suitable example because the it uses freshwater forcing, not global warming, as its control parameter, and (b) if it would be feasible to estimate alpha from quasi-equilibrium experiments with state-of-the-art models (e.g., Garbe et al. 2020; van Westen et al. 2024). These simulations do not need to be analyzed, but a brief comment on the theoretical feasibility on deriving alpha from quasi-equilibrium experiments (instead of knowing the full bifurcation diagram in equilibrium) would be very helpful for the community.

We have realised that equation (5) in the SM (repeated here):

$$\tau_i \frac{\mathrm{d}x_i}{\mathrm{d}t} = -\frac{4\alpha_i^3}{27\beta_i^2 \Delta T_{\mathrm{crit,i}}^2} x_i^3 + \alpha_i x_i + \beta_i \Delta T, \tag{1}$$

is over parameterised. This can be seen by rewriting the equation as:

$$\tau_i \frac{\mathrm{d}x_i}{\mathrm{d}t} = -\frac{4\alpha_i}{27\gamma_i^2 \Delta T_{\mathrm{crit,i}}^2} x_i^3 + \alpha_i x_i + \alpha_i \gamma_i \Delta T, \tag{2}$$

where we have defined $\beta_i/\alpha_i =: \gamma_i$. On the right hand side, a single $\alpha_i$ term appears and therefore, changing $\alpha_i$ would effectively change the timescale $\tau_i$. Hence, without loss of generality, we can set $\alpha_i = 1$. It is important to note that the parameter $\gamma_i$ does not feature in the inverse square law (equation (8) in SM) and thus its choice does not impact the allowed overshoot duration. We have updated the SM accordingly with the reformulation of the equation now included in the SM as equation (6).

**Specific comments:**

It would be good if the concern of reviewer 2 that simply counting the number of tipped elements will not give a representative picture in terms of impacts could be explicitly acknowledged somewhere in the article (even though it is impossible to quantify exactly).

On the first mention of the figure and as we count the number of tipped elements, we now also note that "... the impacts of tipping are very heterogeneous for different Earth system elements."

L20: "and so if . . . ": break this sentence and start a new one with "If . . . ". Otherwise this sentence might suggest a link between the $2.7 \pm 0.2$°C by 2100 and 4°C stabilization, but the Climate Action Tracker does

not assess stabilization levels.

We have split the sentence into two as suggested.

L21: "very unlikely" is this following IPCC-calibrated language?

Yes, it was following the IPCC terminology. However, for clarity, we have decided to rephrase the sentence to highlight that these two elements are the only elements to "have the lower bound of their tipping thresholds above 4°C."

L24: Use "e.g." with this reference

Added as suggested.

L45: Remove comma after "3°C"

The comma has been removed.

L63: It would be good to add a comment after this sentence that the feasibility cannot be assessed in this study.

We have now added "However, assessing the limits of feasibility is outside the scope of this study."

L66: "threshold values may have been determined" → "threshold values of some tipping elements may have been determined" (you mention in your replies that for ice sheets, for example, they appear to be equilibrium values)

Changed as suggested.

L71: Following the discussion in the first round of reviews, it seems appropriate to replace "conventional wisdom" with a more neutral term.

Following the other reviewer's suggestion this has been changed to "frequent assumption".

L73: "all tipping elements could be avoided" → "tipping of all elements considered here could be avoided"

Changed as suggested.

Video supplement: It would be great to have a version that includes the uncertainty assessment. Also, please deposit this in a repository for more permanent archival (github does not guarantee that).

We have decided not to include an additional animation. However, we further emphasise that the animation we do provide uses the best estimates. We have now given the video supplement a fixed DOI by linking the github repository to Zenodo.

The degree signs look like italic "o"s, please use the actual degree sign (° or °) throughout.

This typo has now been corrected throughout the manuscript.

Supplementary Material: I would suggest adding a table or figure which shows the values (or distributions) used from the Armstrong McKay study. Otherwise, the reader needs to flip between three different documents (this article, the supplementary material and the Armstrong McKey et al. paper) to follow along.

A table has now been added to show the values used from the Armstrong McKay study.

check that all mathematical symbols are defined explicitly. It is also unclear why a new scaling parameter beta is introduced and what it scales.

We have ensured that all mathematical symbols are defined explicitly. The parameter $\beta$ (and later $\gamma$) is required to ensure that the units agree. We add specifically "that $[\beta]$ scales between temperature and the units of the state variable $x$". Please note that it was the parameter $\alpha$ that was not required, because as explained above, changing this effectively changes the tipping timescale.

SM L50: could you briefly justify the choice of an exponentially modified Gaussian distribution?

This is one of the simplest distributions that can "accommodate skewed distributions".

SM L52: "no estimate is given" To how many elements does this apply?

We now explicitly state that these refer to the timescales of the coral reefs, East Antarctic ice sheet, Barents sea ice and boreal forests. Based on this request, we now write in the papr: "Note, for the coral reefs, East Antarctic ice sheet, and the Barents sea ice only, one timescale estimate is given and so these are assumed to have no uncertainty. Similarly, the upper estimate for the timescales of the boreal forests are unknown and so the upper estimate takes the value of the best estimate for both dieback and expansion respectively, these modifications are included in Table 1."

Garbe et al. 2020, https://www.nature.com/articles/s41586-020-2727-5
Van Westen et al. 2024, https://www.science.org/doi/10.1126/sciadv.adk1189

**Response to Reviewer 3**

We are pleased to hear that we addressed all of the reviewer's comments and that there are only some minor comments remaining. These comments are repeated in black, and our responses are given in blue.

The authors addressed all of my comments. I have some minor remarks remaining, mostly regarding phrasing.

L.24: In accordance with the authors' response to my previous comment (previously L.1), I suggest either expanding the sentence or adding another short sentence that addresses the framing of tipping points in the media as instantaneous events.

We have now added the following sentence to address the framing of tipping points in the media as instantaneous events: "In the media it is often incorrectly implied that tipping is instantaneous upon crossing the threshold (e.g. Guardian article: "World on brink of five 'disastrous' climate tipping points", September, 2022)."

L.71: I still do not like the phrase "conventional wisdom" here, as it seems to overlook some of the progress made in recent years. I understand what the authors mean, but I would recommend rephrasing it to something like "frequent assumption" or a similar expression.

We have rephrased as suggested.